# Hybridization of short-range and long-range charge transfer excited states in multiple resonance emitter

Ha Lim Lee[1,2,5], Jihoon Kang[1,5], Junseop Lim[1], Seung Chan Kim[1], Soon Ok Jeon [2] ✉ & Jun Yeob Lee [1,3,4] ✉

Multiple resonance (MR) thermally activated delayed fluorescence emitters have been actively studied as pure blue dopants for organic light-emitting diodes (OLEDs) because of excellent color purity and high efficiency. However, the reported MR emitter, 2,5,13,16-tetra-*tert*-butylindolo[3,2,1-*jk*]indolo[1′,2′,3′:1,7]indolo[2,3-*b*]carbazole (tDIDCz) based on bis-fused indolo-carbazole framework could not demonstrate efficient triplet-to-singlet spin crossover. In this work, we report two isomeric MR emitters designed to promote triplet exciton harvesting by reconstructing the electronic structure of tDIDCz. To manage excited states, strong electron donors were introduced at the 2,5-/1,6-position of tDIDCz. As a result, 2,5-positions managed tDIDCz shows long-range charge transfer characteristics while preserving the MR nature. Quantum chemical calculation demonstrates direct spin-orbit coupling by long-range charge transfer and spin-vibronic coupling assisted reverse intersystem crossing by short-range charge transfer simultaneously contribute to triplet-to-singlet spin crossover. Consequently, high performance blue OLED recorded a high external quantum efficiency of 30.8% at a color coordinate of (0.13, 0.13).

Since the first developments of organic light-emitting diodes (OLED) using fluorescent emitters in 1987[1], a lot of efforts have been devoted to upgrade the efficiency and lifetime by exploiting a mechanism fully utilizing excitons for radiative transition[2–8]. Among the approaches, pure organic based thermally activated delayed fluorescence (TADF) emitters have been spotlighted as an alternative to highly efficient phosphorescent emitters utilizing expensive noble metals. Internal quantum efficiency of 100% could be realized in the TADF devices by harvesting non-radiative triplet excitons into radiative singlet excitons via reverse intersystem crossing (RISC)[9–11].

Numerous TADF emitters have been designed based on the donor–acceptor (D–A) building blocks exhibiting charge transfer (CT) character by localization of the highest occupied molecular orbital (HOMO) on donor moieties and the lowest unoccupied molecular orbital (LUMO) on acceptor moieties[12–15]. Extensive separation of frontier molecular orbitals (FMOs) could make the singlet-triplet energy splitting ($\Delta E_{ST}$) smaller and the rate constant of RISC process ($k_{RISC}$) higher. However, the strong CT character makes the emission band broader, degrading the color purity of the D-A type TADF emitters[16,17]. To solve the above-mentioned issue, a multiple resonance (MR) type TADF design strategy was suggested and firstly reported in 2016[18–20]. The MR–TADF emitters have electron-rich atoms and electron-deficient atoms coexisting in a fused structure called polycyclic aromatic hydrocarbon (PAH). Alternating HOMO and LUMO

[1]School of Chemical Engineering, Sungkyunkwan University 2066, Seobu-ro, Jangan-gu, Suwon-si, Gyeonggi-do 16419, Korea. [2]Samsung Advanced Institute of Technology, Samsung Electronics, 130 Samsung-ro, Suwon, Gyeonggi 16678, Republic of Korea. [3]SKKU Advanced Institute of Nano Technology, Sungkyunkwan University 2066, Seobu-ro, Jangan-gu, Suwon, Gyeonggi 16419, Republic of Korea. [4]SKKU Institute of Energy Science and Technology, Sungkyunkwan University 2066, Seobu-ro, Jangan-gu, Suwon, Gyeonggi 16419, Republic of Korea. [5]These authors contributed equally: Ha Lim Lee, Jihoon Kang. ✉e-mail: so.jeon@samsung.com; leej17@skku.edu

distributions are observed within short range, which is favoured for efficient radiative transition and local emission character. Therefore, the MR–TADF emitters derived from boron and nitrogen show narrow emission with a full width at half maximum (FWHM) of below 30 nm by a rigid $\pi$-conjugated structure and local emission nature, and high photoluminescence quantum yield (PLQY), resulting in good color purity and high efficiency in the devices. However, the RISC process was rather slow because of large $\Delta E_{ST}$.

Recently, our group reported new MR emitters derived from indolo[3,2,1-*jk*]carbazole (ICz) moiety. The fusion of two ICz units produced the emitter namely 2,5,13,16-tetra-*tert*-butylindolo[3,2,1-*jk*]indolo[1′,2′,3′:1,7]indolo[2,3-*b*]carbazole (tDIDCz) that showed small Stokes shift of 3 nm and small FWHM of 14 nm owing to rigid structure with MR character[21]. However, the emission of tDIDCz exhibiting in the purple color region led to a large $\Delta E_{ST}$ of 0.44 eV. Therefore, it could not present any TADF characteristics in spite of narrow emission spectrum, affording a low external quantum efficiency (EQE) under 5%. The low EQE hurdle of the previous work might be handled by managing the excited state for small $\Delta E_{ST}$ while keeping the MR properties.

Herein, we describe two emitters, 13,16-di-*tert*-butyl-$N^2,N^2,N^5,N^5$-tetrakis(4-(*tert*-butyl)phenyl)indolo[3,2,1-*jk*]indolo[1′,2′,3′:1,7]indolo[2,3-*b*]carbazole-2,5-diamine (2,5-tDPAtDIDCz) and 13,16-di-*tert*-butyl-$N^1,N^1,N^6,N^6$-tetrakis(4-(*tert*-butyl)phenyl)indolo[3,2,1-*jk*]indolo[1′,2′,3′:1,7]indolo[2,3-*b*]carbazole-1,6-diamine (1,6-tDPAtDIDCz), with two di(*tert*-butylphenyl)amine (tDPA) groups at 2,5- positions or 1,6- positions of tDIDCz as the excited state manager adjusting the emission energy and origin of excited states. Different effects of donor substitution position (2,5- or 1,6- position) on tDIDCz core provided formation of significantly different electronic structures. The excited state manager red-shifted the emission spectrum from violet to deep blue region and opened the RISC channel for TADF emission by hybridizing long-range CT (LRCT) and short-range CT (SRCT) excited state character. It was demonstrated that direct RISC by LRCT and spin-vibronic coupling (SVC) assisted RISC by SRCT through MR core structure simultaneously contributed to the up-conversion of triplet excitons. Among the two emitters, the 2,5-tDPAtDIDCz showed high EQE of 30.8% by TADF emission and small FWHM of 38 nm with y color coordinate of 0.13. This work proposed that the hybridization of LRCT and SRCT excited states can realize both high EQE and narrow emission spectrum.

## Results

### Molecular design strategy

The hybridization concept of LRCT and SRCT excited states is to utilize the narrow emission from MR character and efficient RISC from CT character as presented in Fig. 1. The CT nature of the excited state would reduce $\Delta E_{ST}$ of the MR emitter for RISC process to harvest the triplet excitons for fluorescence. As a macroscopic HOMO–LUMO

separation is essential for the CT excited state character, an electron rich tDPA was introduced as an excited state manager to the MR type tDIDCz core which can be regarded as an electron poor MR structure. The tDPA substituent may hybridize the excited states by CT properties of the molecule through D–A nature and MR properties of the tDIDCz core. Therefore, simultaneous achievement of narrow emission and efficient TADF emission is anticipated from the tDPA substituted tDIDCz emitters. Additionally, the tDPA units play a role of preventing intermolecular interaction by molecular packing.

The substitution positions of the tDPA excited state manager in the tDIDCz core were 2,5- or 1,6- positions. Two substitution approaches were chosen because of different orbital distribution at the two positions. The 1,6- positions of tDIDCz are nodes for the HOMO, while the 2,5- positions are electron rich region (Supplementary Fig. 1). The donor substitution at the nodes of HOMO may weakly extend the HOMO for CT properties, while that at the HOMO rich part may significantly extend the HOMO for strengthened CT properties. The different substitution positions may induce large discrepancy of the molecular orbital distribution and photophysical properties by managing the singlet and triplet excited states. Therefore, 2,5-tDPAtDIDCz and 1,6-tDPAtDIDCz were designed and synthesized as the blue emitters derived from the tDIDCz MR core.

The synthetic pathway of 2,5-tDPAtDIDCz and 1,6-tDPAtDIDCz and its molecular structure are described in Fig. 2 and Supplementary Fig. 2. The detailed synthetic procedures and analytic information are provided Supplementary Note section. The same synthetic procedure was employed in the synthesis of the emitters except for the tDPA modified carbazole intermediates. Asymmetric substitution of *tert*-butyl modified carbazole and tDPA modified carbazole was carried out by stepwise reaction of the two intermediates followed by final ring closing reaction.

### Quantum chemical calculation

Time-dependent density functional theory (TD-DFT) calculations were performed on ground state geometries using the B3LYP[22] functional and the 6-31 G(d,p) basis set in gas phase. Vertical excited-state geometries and energies were derived by TD-DFT at the MPW1B95[23]/6-31 G(d,p) level of theory. The spin-orbit coupling (SOC) matrix element estimation between the $S_1$ and $T_n$ states were carried out using ground state geometries and the optimized ground state three dimensional coordinates are provided in Supplementary Table 1. All the DFT and TD-DFT computations were performed using Gaussian 09 package[24], and the SOC calculations were performed using ADF package[25]. Fig. 3 provides the excited state characteristics and emission mechanism of the two deep blue emitters analyzed by quantum chemical calculation. In addition, the highest occupied NTO (HONTO) and the lowest unoccupied NTO (LUNTO) distributions of 2,5-tDPAtDIDCz and 1,6-tDPAtDIDCz at various excited states are illustrated in Supplementary

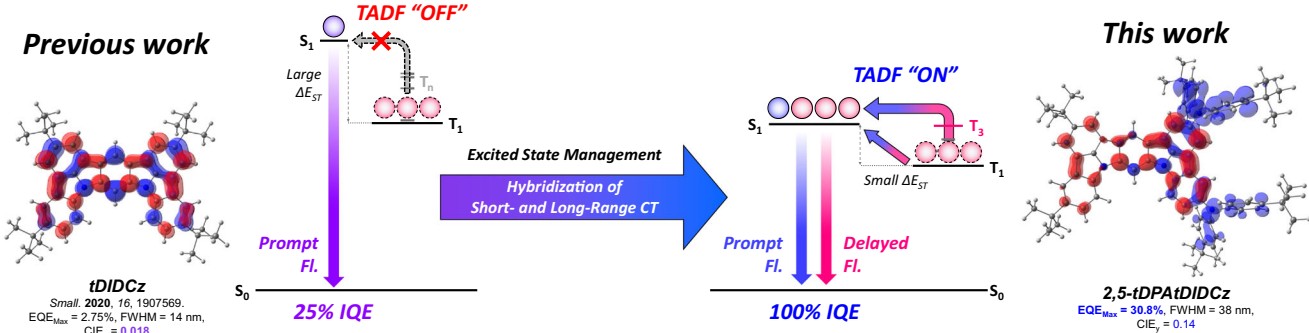

**Fig. 1 | Molecular design strategy.** Schematic illustration of hybridization of short- and long-range charge transfer excited state applied to 2,5-tDPAtDIDCz (right). The electronic structure of tDIDCz (left) is reorganized by adopting bis(4-(*tert*-butyl) phenyl)amine units. The blue/red colored parts are HOMO/LUMO distribution of corresponding molecules, respectively.

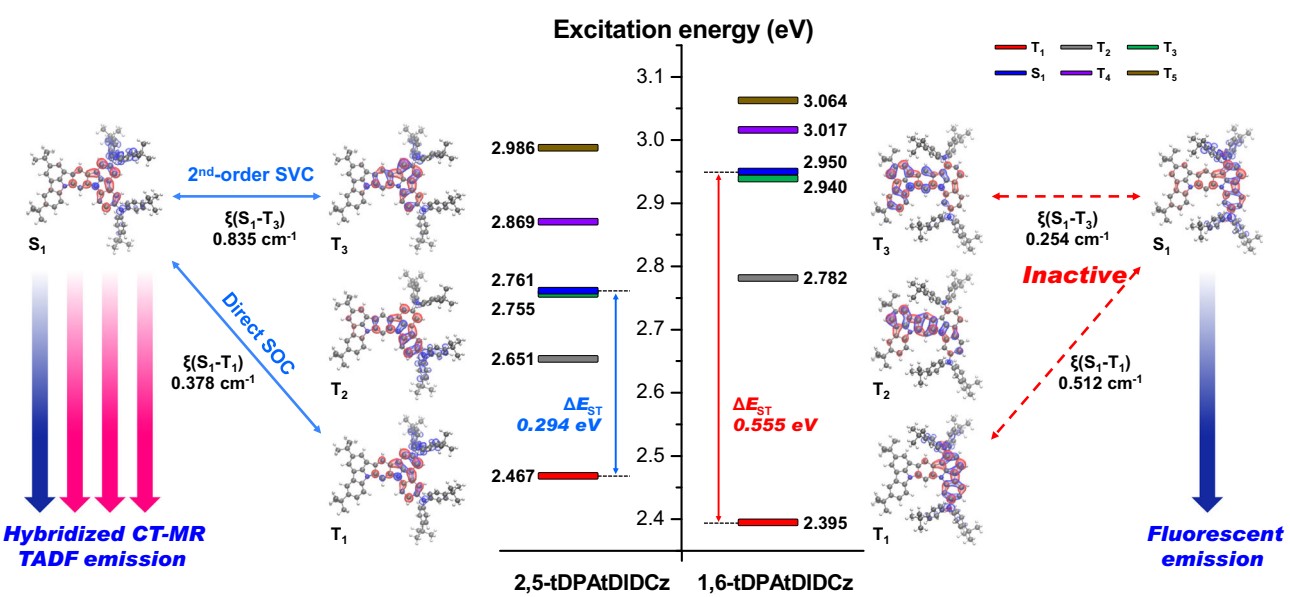

**Fig. 2 | Synthesis and structures.** Overall synthetic scheme and molecular structures of two multiple-resonance emitters.

**Fig. 3 | Electronic structure calculation.** The results of natural transition orbital calculation and the mechanism of hybridized charge transfer (CT)–multiple resonance (MR) thermally activated delayed fluorescence (TADF) emission.

Fig. 3 and Supplementary Fig. 4. As confirmed in the FMO distributions, a long-range charge separation was induced in the tDIDCz derivatives at singlet excited state ($S_1$) due to the introduction of tDPA excited state manager. The stabilized $S_1$ energies of 2,5-tDPAtDIDCz and 1,6-tDPAtDIDCz were 2.761 and 2.950 eV, respectively. In the NTO distribution of both molecules, it was observed that the electron accepting tDIDCz core and electron donating tDPA are dominantly occupied by electron and hole, respectively, while partially maintaining the MR distribution of the core. From the HONTO–LUNTO distributions, it was confirmed that the CT and MR characteristics coexist in the $S_1$, forming a hybridized excited state. The CT contribution to the excited state was relatively large in the 2,5-tDPAtDIDCz, judging from the degree of LUNTO dispersion through the MR structure. Although the MR and CT characteristics are hybridized in the two emitters, the MR character is more dominant in the 1,6-tDPAtDIDCz than 2,5-tDPAtDIDCz. Therefore, the $S_1$ energy of 2,5-tDPAtDIDCz was

significantly lowered by LRCT interaction between *para*-oriented sp² nitrogen of ICz and sp³ nitrogen of tDPA. Whereas, the $S_1$ energy of 1,6-tDPAtDIDCz was moderately stabilized by linearly extended conjugation between ICz and two tDPA units. The triplet excited state ($T_1$) of the two emitters was also observed to have a hybridized excited state character similar to that of $S_1$. The calculated $T_1$ energies of 2,5-tDPAtDIDCz and 1,6-tDPAtDIDCz were 2.467 and 2.395 eV, respectively. The extended conjugation induced the low $T_1$ energy of 1,6-tDPAtDIDCz. As a result, 2,5-tDPAtDIDCz showed small $\Delta E_{ST}$ of 0.294 eV compared to 0.555 eV of 1,6-tDPAtDIDCz. The dramatically reduced $\Delta E_{ST}$ of 2,5-tDPAtDIDCz suggests that direct SOC-induced $T_1$ to $S_1$ spin crossover can be opened. In addition, the $T_3$ state of 2,5-tDPAtDIDCz had an excitation energy similar to that of the $S_1$ state and it is predicted to be a LE dominant state because HONTO and LUNTO are distributed in DIDCz core. This implies that the close-lying $T_3$ state is suitable as the SVC-mediating high–lying $T_n$ state for CT dominant $S_1$

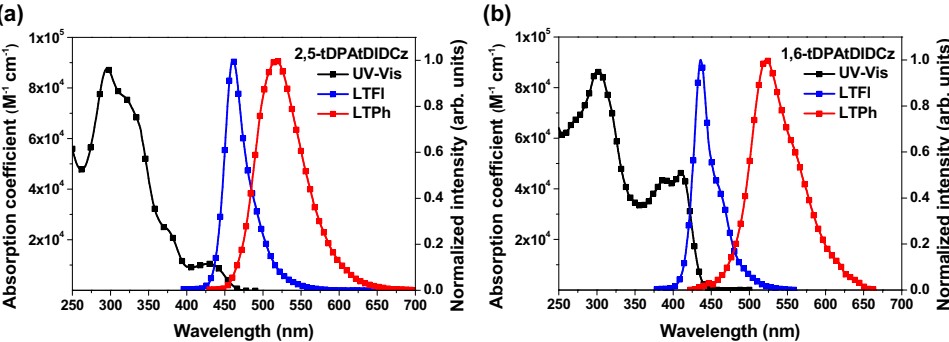

**Fig. 4 | Absorption and emission spectra.** The ultraviolet-visible (UV-Vis) absorption and photoluminescence emission spectra of (**a**) 2,5-tDPAtDIDCz and (**b**) 1,6-tDPAtDIDCz. LTFl represents low temperature fluorescence and LTPh represents low temperature phosphorescence.

state to induce efficient RISC. The NTO calculation results propose that the tDPA could control the nature of the excited states as the excited state manager.

To analyze the triplet exciton harvesting mechanism, the SOC matrix element (SOCME) between different excited states was calculated. Supplementary Table 2 summarizes the excitation energy gap and SOCME between different excited states. The origin of TADF emission of 2,5-tDPAtDIDCz can be regarded as (i) $T_1 \rightarrow S_1$ up-conversion by direct spin-orbit coupling (SOC) and (ii) SVC assisted RISC processes via SOC interaction between $T_n$ and $S_1$. Here, direct SOC is activated by long range CT as commonly observed in D-A type structures[26,27] and the SVC-RISC mechanism is enabled by MR structure possessing short range CT character[28–30]. In the SVC-RISC model, it was explained that the $T_1$ to $S_1$ up-conversion pathway is theoretically possible via the high lying $T_n$ showing strong SOC interaction with $S_1$ in spite of large energy difference higher than 0.2 eV between $S_1$ and $T_1$. It has been known that the $2^{nd}$-order SVC-RISC process occurs effectively only when vibrational coupling between $T_1$ and high lying $T_n$ is efficient and the gap between $\Delta E_{ST}$ and triplet-triplet energy splitting ($\Delta E_{TT}$) is quite small[29]. In the case of 2,5-tDPAtDIDCz, the difference between $\Delta E_{ST}$ and $\Delta E_{TT}$ is only 0.006 eV and the SOCME for the $S_1$-$T_3$ is as large as 0.835 cm$^{-1}$, indicating efficient RISC by SVC mechanism. Additionally, the small $\Delta E_{ST}$ and large SOCME of 0.378 cm$^{-1}$ between $S_1$ and $T_1$ trigger the direct up-conversion from $T_1$ and $S_1$. In the case of 1,6-tDPAtDIDCz, small SOCME of 0.254 cm$^{-1}$ for SVC process and large $\Delta E_{ST}$ of 0.555 eV for direct spin conversion due to weak CT character prohibit the triplet exciton harvesting via RISC. In summary, 2,5-tDPAtDIDCz exhibited sufficiently small $\Delta E_{ST}$ due to the introduction of the excited state manager, and the reconstructed triplet excited states provided appropriate NTO distributions and SOCME values suitable for harvesting triplet exciton via direct SOC and $2^{nd}$-order RISC. On the other hand, as can be seen in 1,6-tDPAtDIDCz, the linearly extended arrangement of two tDPA units could not build proper electronic structure suitable for triplet exciton harvesting. Therefore, 2,5-tDPAtDIDCz may show TADF emission, while 1,6-tDPAtDIDCz may exhibit conventional fluorescence.

**Material characterization**

The NTO analysis predicted CT and MR hybridized excited state for the 2,5-tDPAtDIDCz and 1,6-tDPAtDIDCz. The calculation results were experimentally validated by analysing the photophysical transitions of the emitters. The electronic transitions corresponding to absorption and emission were recorded using ultraviolet-visible (UV-Vis) and photoluminescence (PL) measurements. The collected absorption and emission spectra are presented in Fig. 4. The UV-Vis absorption pattern of the two emitters was similar at below 350 nm due to local absorption from the same core structure, but different absorption profiles were observed between 350 and 470 nm. Weak CT absorption peak was discerned at 430 nm in the 2,5-tDPAtDIDCz, while strong local

absorption bands were detected between 350 and 450 nm in the 1,6-tDPAtDIDCz. The significant extension of the conjugated structure by *para*-orientation of the linearly extended two tDPA units gave rise to the local absorption at long wavelength in the 1,6-tDPAtDIDCz. The *meta*-oriented tDPA units in the 2,5-tDPAtDIDCz inhibited the conjugation extension. Instead, it allowed CT absorption by electronic isolation of the excited state managing tDPA units.

The radiative transitions represented by fluorescence and phosphorescence can be correlated with the origin of the singlet and triplet excited states. Comparing the fluorescence by singlet excitons of the two emitters, local emission character was relatively strong in the 1,6-tDPAtDIDCz emitter judging from the sharp emission spectrum and clear vibrational peak. The slightly broadened spectrum and weak shoulder in the 2,5-tDPAtDIDCz propose combined emissions from LRCT and SRCT excited states. The FWHM of the fluorescence spectrum of 2,5-tDPAtDIDCz was only 32 nm compared to 29 nm of 1,6-tDPAtDIDCz, demonstrating the unique feature of the MR emission in spite of appended CT property. The CT character of the two emitters was confirmed by the solvatochromic effect measured at room temperature (Supplementary Fig. 5). The peak wavelength of 2,5-tDPAtDIDCz was 455 nm in *n*-hexane and 478 nm in methylene chloride, showing a red-shift of 23 nm, while 1,6-tDPAtDIDCz showed red-shift of 18 nm, indicating that 2,5-tDPAtDIDCz possesses relatively strong CT character. The phosphorescence spectra of the two emitters also reflected similar excited state nature even in triplet excited states. More CT character was embedded in the 2,5-tDPAtDIDCz. The onset wavelengths of fluorescence/phosphorescence were 438/468 nm and 415/479 nm in the 2,5-tDPAtDIDCz and 1,6-tDPAtDIDCz, respectively. Red-shift of fluorescence and blue shift of phosphorescence in the 2,5-tDPAtDIDCz emitter are due to intensified CT character and disconnected conjugation structure, respectively. As a result, the $\Delta E_{ST}$ values of 2,5-tDPAtDIDCz and 1,6-tDPAtDIDCz were 0.18 and 0.41 eV, respectively. Considering the $\Delta E_{ST}$ of the emitters, 2,5-tDPAtDIDCz may work as a TADF emitter, while 1,6-tDPAtDIDCz may serve as a conventional fluorescent emitter.

The analysis of electrochemical property of two emitters was proceeded by measuring oxidation and reduction curves from cyclic voltammetry (Supplementary Fig. 6). The HOMO and LUMO energy levels were −5.55/−2.74 eV for 2,5-tDPAtDIDCz and −5.54/−2.66 eV for 1,6-tDPAtDIDCz. Both emitters showed shallow HOMO energy level and reduced HOMO-LUMO gap compared with those of tDIDCz (−5.98/−2.59 eV), because of additional tDPA substituents possessing strong electron donating nature. The summarized material characterization data are presented in Table 1.

The TADF nature of the emitters was substantiated by tracing a slowly decaying delay fluorescence using transient PL (TRPL) analysis. Vacuum deposited films of 2,5-tDPAtDIDCz and 1,6-tDPAtDIDCz were prepared in 1,3-di(9H-carbazol-9-yl)benzene (mCP): diphenyl(4-(triphenylsilyl)phenyl)phosphine oxide (TSPO1) blended matrix at 1 wt%

**Table 1 | Summarized material characterization data**

| | HOMO[a] (eV) | LUMO[a] (eV) | $\lambda_{abs}$[b] (nm) | $\lambda_{em}$[c] (nm) | $E_S$[c] (eV) | $E_T$[c] (eV) | $\Delta E_{ST}$[c] (eV) | Stokes Shift (nm) | FWHM (nm) |
|---|---|---|---|---|---|---|---|---|---|
| **2,5-tDPAtDIDCz** | −5.55 | −2.74 | 429 | 438 | 2.83 | 2.65 | 0.18 | 32 | 32 |
| **1,6-tDPAtDIDCz** | −5.54 | −2.66 | 411 | 415 | 2.99 | 2.58 | 0.41 | 25 | 29 |

[a]measured by oxdiation and reduction method using 0.1 M of tetrabutylammonium perchlorate in acetonitrile solution with ferrocene and mCP as a reference material. [b]measured at room temperature with $10^{-5}$ M of THF solution. [c]measured at 77 K in frozen THF matrix without and with 1.0 millisecond delay.

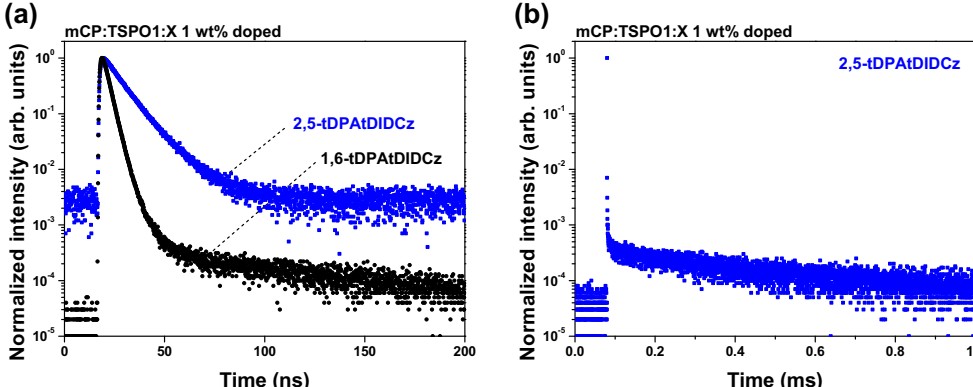

**Fig. 5 | Transient photoluminescence decay curves. a** Prompt component decay curves of 2,5-tDPAtDIDCz (blue) and 1,6-tDPAtDIDCz (black). **b** Delayed component decay curve of 2,5-tDPAtDIDCz (blue).

doping concentration. The prompt and delay components of the TRPL measurements in Fig. 5 validate the TADF property of 2,5-tDPAtDIDCz and 1,6-tDPAtDIDCz. The 1,6-tDPAtDIDCz emitter exhibited only prompt fluorescence without delayed fluorescence, while the 2,5-tDPAtDIDCz emitter showed both prompt and delayed fluorescence. The tDIDCz emitter was a conventional fluorescence emitter without any TADF emission, but it was transformed into a TADF emitter by modifying 2 and 5-positions of tDIDCz with tDPA, whereas 1 and 6-positions modified tDIDCz did not exhibit TADF characteristic (Supplementary Fig. 7). This result is well correlated with the NTO simulation results and photophysical measurement data of the two emitters. The photoluminescence quantum yields (PLQYs) of 2,5-tDPAtDIDCz and 1,6-tDPAtDIDCz were 88 and 92%, respectively, indicating efficient radiative transition assisted by the MR core structure. The added CT character in the 2,5-tDPAtDIDCz emitter through the tDPA excited state manager did not largely degrade the PLQY. The rate constant for RISC from PLQY and excited state lifetime was $1.28 \times 10^4\,\mathrm{s}^{-1}$. Radiative transition related parameters of the two emitters are summarized in Supplementary Table 3.

**Device fabrication and evaluation**

A triplet exciton harvesting device structure was employed to evaluate electroluminescence (EL) performances of the emitters. The device structure with energy levels of the materials is provided in Fig. 6(a). The device performance data of blue OLEDs including EL spectra, current density–voltage–luminance curves, and EQE–current density curves are illustrated in Fig. 6(b–d). The EQEs of the 2,5-tDPAtDIDCz and 1,6-tDPAtDIDCz devices reflected the TADF characteristics of the emitters by providing maximum EQEs of 23.4 and 6.4%, respectively. The low EQE under 5% of tDIDCz was significantly upgraded to over 20% by substitution of tDPA at 2 and 5 positions of tDIDCz for TADF properties. The absence of TADF character resulted in the low EQE of 6.4% in the 1,6-tDPAtDIDCz. However, the EQE was dramatically enhanced compared to that of tDIDCz by the two tDPA units. The EQE of the 2,5-tDPAtDIDCz device was further enhanced by replacing the mCP:TSPO1 host with 2,6-bis(3-(9H-carbazol-9-yl)phenoxy)benzonitrile (mBisPCz-O-BN) for improved carrier balance and horizontal

dipole orientation (85% at mCP:TSPO1 mixed host, 93% at mBisPCz-O-BN bipolar host, Supplementary Fig. 8), enabling high EQE of 30.8%. The device structure and device characteristic curves of further optimized blue devices are illustrated in Supplementary Fig. 9. However, the mBisPCz-O-BN hosted 1,6-tDPAtDIDCz device recorded EQE_Max of 7.3%. Although the EQE was slightly improved, there was no significant difference compared to that of the mCP:TSPO1 hosted device. The host matrix did not activate the SVC-RISC process of 1,6-tDPAtDIDCz, and 1,6-tDPAtDIDCz device still showed only fluorescence without TADF process. The EQE–luminance and power efficiency–luminance curves of mCP:TSPO1 hosted devices and mBisPCz-O-BN hosted devices are illustrated in Supplementary Fig. 11 and Supplementary Fig. 12. To understand the origin of improved EQE in the mBisPCz-O-BN hosted device, transient PL and PLQY analysis of the emitter doped mBisPCz-O-BN films was conducted (Supplementary Fig. 13). The PL analysis revealed that mBisPCz-O-BN:2,5-tDPAtDIDCz film provided an improved PLQY of 92% and significantly lowered $k_{ISC}$ while maintaining $k_{RISC}$. Therefore, it can be concluded that the mBisPCz-O-BN host dramatically improved the EQE of the 2,5-tDPAtDIDCz device by enhancing the outcoupling efficiency and PLQY compared to the mCP:TSPO1 host. Supplementary Fig. 14 provides the device stability measured under a constant current density condition at an initial luminance of 100 cd m$^{-2}$. All devices exhibited half device lifetime of less than 3 h, which presumed to be due to poor material stability of host and charge transport layer.

The 2,5-tDPAtDIDCz device still maintained the narrow emission spectrum while dramatically increasing the EQE. The FWHMs of the 2,5-tDPAtDIDCz device with mCP:TSPO1 and mBisPCz-O-BN hosts were 36 and 38 nm, respectively. In spite of the appended CT property, sharp emission profile was demonstrated due to the rigid MR core structure taking part in the excited state. The 1,6-tDPAtDIDCz showed even narrower emission band with FWHM of 32 nm. The color coordinates of the 2,5-tDPAtDIDCz and 1,6-tDPAtDIDCz devices were (0.13, 0.12) and (0.16, 0.04), respectively. The participation of the rigid MR chromophore for fluorescence sharpened the emission spectrum while achieving high EQE by TADF through combined CT properties. Table 2 provides all device performances reported in this work, and

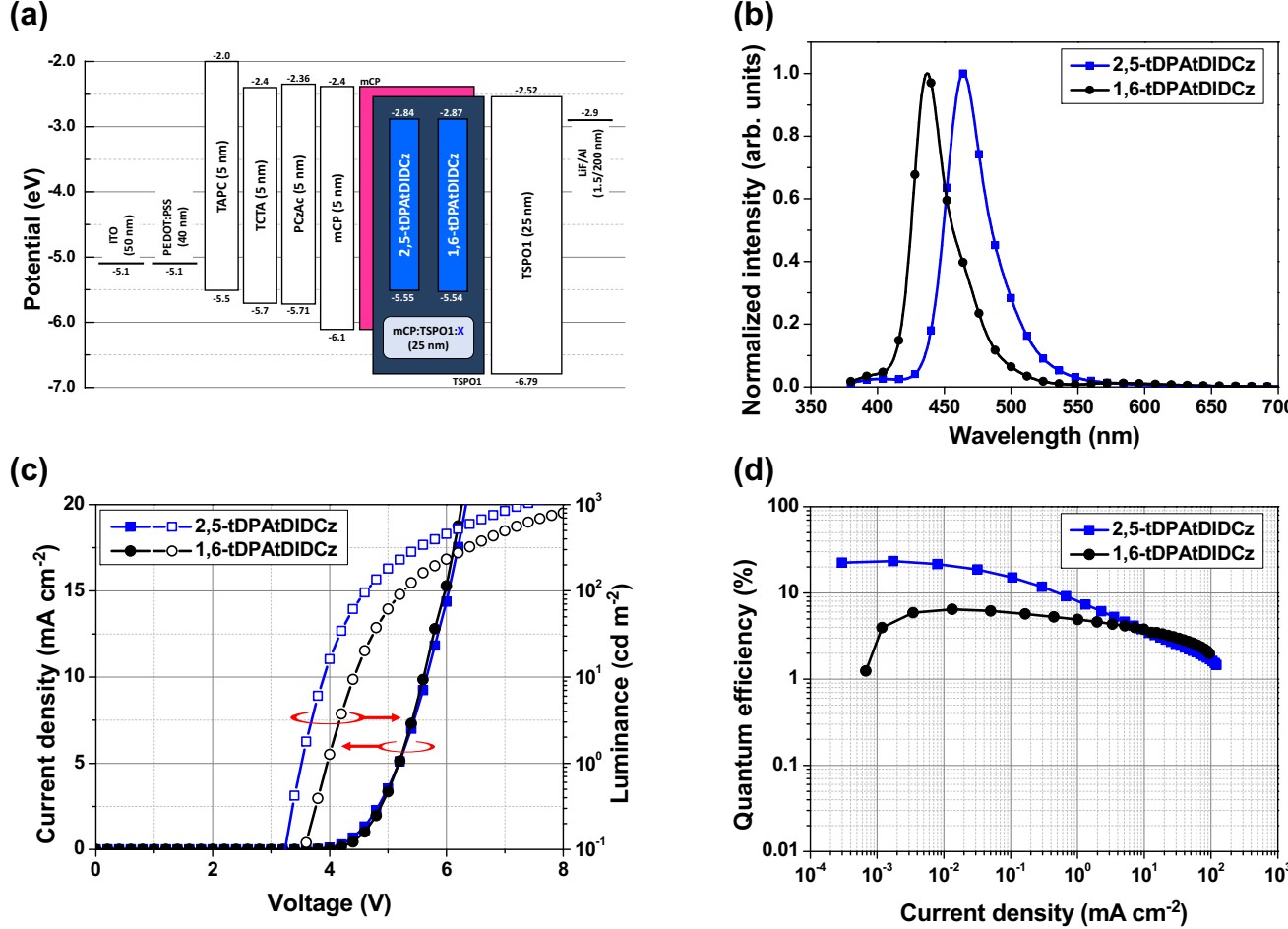

**Fig. 6 | OLED device performance. a** The device structure and energy level diagram of mCP:TSPO1 hosted blue OLEDs. **b** Normalized electroluminescence spectra. **c** Current density–voltage–luminance curves. **d** External quantum efficiency–current density curves of 2,5-tDPAtDIDCz (blue) and 1,6-tDPAtDIDCz (black) devices.

**Table 2 | Summarized device performance of the blue OLEDs**

| Compound | | $V_{on}$[a] (V) | $V_d$[b] (V) | $\lambda_{EL}$[b] (nm) | FWHM[b] (nm) | CIE (x, y)[b] | EQE[c] (%) | PE[c] (lm W$^{-1}$) | CE[c] (cd A$^{-1}$) |
|---|---|---|---|---|---|---|---|---|---|
| **2,5-tDPAtDIDCz** | 1%[d] | 3.7 | 4.7 | 464 | 36 | (0.13, 0.12) | 23.4 | 23.1 | 24.2 |
| | 1%[e] | 3.5 | 4.8 | 466 | 38 | (0.13, 0.14) | 30.8 | 31.5 | 34.1 |
| **1,6-tDPAtDIDCz** | 1%[d] | 4.1 | 5.4 | 437 | 32 | (0.16, 0.04) | 6.4 | 3.1 | 3.4 |
| | 1%[e] | 3.9 | 6.1 | 438 | 36 | (0.16, 0.05) | 7.3 | 8.8 | 8.7 |

[a]Turn-on voltage at a luminance of 1 cd m$^{-2}$. [b]Driving voltage at a luminance of 100 cd m$^{-2}$. [c]Maximum values. [d]1 wt% doped mCP:TSPO1 hosted device. [e]1 wt% doped mBisPCz-O-BN hosted device.

Supplementary Table 4 provides the summarized device performance of the reported blue OLEDs.

In this work, deep-blue MR type emitters were developed by managing the excited states of tDIDCz using a tDPA excited state manager which enables hybridized LRCT and SRCT excited state. Efficient reconstruction of excited states by substituting the excited state manager at 2 and 5 position of tDIDCz allowed triplet exciton harvesting through two RISC processes of direct RISC from $T_1$ to $S_1$ by strong SOC and second-order SVC mediated RISC from $T_3$ to $S_1$ by MR character. As a result of hybridization of LRCT and SRCT excited state, 2,5-tDPAtDIDCz exhibited a small FWHM of 32 nm, high PLQY of 88%, and a delayed decay time of 251.4 μs. Consequently, the 2,5-tDPAtDIDCz devices achieved a high EQE of 30.8%, small FWHM of 38 nm, and color coordinate of (0.13, 0.13).

## Methods
### General information
All starting materials, solvents, bases, catalysts were commercially available and used without any purification. The structure analysis of synthesized intermediates and final compounds was performed with elemental analysis (organic elemental analyzer (vario EL cube, Elementar)), nuclear magnetic resonance (NMR) (Oxford 300 NMR (VARIAN, 300 MHz) and Bruker ASCEND 500 device at 500 MHz) and mass spectroscopy (atmospheric pressure chemical ionization (APCI) with Advion, Expresion$^L$ CMS spectrometer and high resolution liquid chromatograph mass spectrometer–ion trap–time-of-flight (LCMS-IT-TOF, Shimadzu)). The detailed synthetic information and analytic information are described in Supplementary Note section in Supplementary Information.

## Material characterization

UV-Vis absorption spectra and PL emission spectra were measured with dilute tetrahydrofuran (THF) solution ($1 \times 10^{-5}$ M), using UV-Vis spectrophotometer (JASCO, V-730) and fluorescence spectrophotometer (PerkinElmer, LS-55), respectively. The fluorescent and phosphorescent spectra were gathered without and with 1 ms delay at 77 K. For solid PL measurement, thin films by vacuum deposition were prepared. The absolute PL quantum yield (PLQY) and transient PL curves with solid film were recorded using Quantaurus-QY system (Hamamatsu, C11347-11) and Quantaurus-Tau system (Hamamatsu, C11367-31) under a nitrogen atmosphere. The doping concentration of deposit film was 1 wt%. The energy level of HOMO and LUMO was estimated with cyclic voltammetry (Ivium Tech., Iviumstat). The oxidation and reduction curves were obtained by using working electrode (glassy carbon tube), counter electrode (platinum wire) and reference electrode (saturated Ag/AgCl). As an electrolyte, 0.1 M of tetrabutylammonium perchlorate in acetonitrile solution was used. As a standard material, ferrocene was used. All the solutions were purged with nitrogen for 10 min to remove oxygen.

## Device fabrication

The bottom emission device structure of blue OLED was indium tin oxide (ITO) (50 nm) / PEDOT:PSS (40 nm) / TAPC (5 nm) / TCTA (5 nm) / PCzAc (5 nm) / mCP (5 nm) / mCP:TSPO1:2,5-tDPAtDIDCz or 1,6-tDPAtDIDCz (25 nm, 50%, 1 wt%) / TSPO1 (25 nm) / LiF (1.5 nm)/Al (200 nm). The optimized device structure was ITO (50 nm) / PEDOT:PSS (40 nm) / TAPC (10 nm) / mCP (10 nm) / mBisPCz-O-BN:2,5-tDPAtDIDCz (25 nm, 1 wt%) / TSPO1 (25 nm) / LiF (1.5 nm)/Al (200 nm). The molecular structure of organic electronic materials used in blue OLED devices are illustrated in Supplementary Fig. 11. The ITO substrates were cleaned by ultrasonication with acetone, isopropylalcohol, chloroform and deionized water and dried in oven. Also, oxygen plasma treatment to ITO substrates was performed to fine-tune the work function of anode. A 40 nm thick PEDOT:PSS film was formed on ITO substrate by spin-coating (30 s at 3200 rpm) followed by annealing at 150 °C for 15 min. Vacuum thermal deposition process was proceeded under a pressure of $2.0 \times 10^{-7}$ Torr and the devices were encapsulated with a glass lid in nitrogen-filled glove box. Device characteristics were recorded by using a CS2000 spectroradiometer and Keithley 2400 for electrical and optical output measurement. PEDOT:PSS stands for poly(3,4-ethylenedioxythiophene)-poly(styrenesulfonate). TAPC is (1,1-bis(4-di-*p*-tolylaminophenyl)cyclohexane), TCTA is tris(4-carbazoyl-9-ylphenyl) amine, and PCzAc is 9,9-dimethyl-10-(9-phenyl-9*H*-carbazol-3-yl)-9,10-dihydroacridine.

## Data availability

The authors declare that all relevant data are included in the Supplementary Information of this paper, along with Source data. Source data are provided with this paper.

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

## Acknowledgements
This work was supported by SAIT and National Research Foundation of Korea (2020R1A2C2100872).

## Author contributions
H.L.L. synthesized materials and prepared original manuscript. J.K. synthesized and analyzed the materials and revised the manuscript. J.L. fabricated devices. S.C.K. evaluated the devices. S.O.J. proposed the design concept of the materials and supervised the project. J.Y.L. prepared final manuscript and supervised the project.

## Competing interests
The authors declare no competing interests.
