## [Peer Review File · Nature Communications]

Hybridization of Short-Range and Long-Range Charge Transfer Excited States in Multiple Resonance EmitterREVIEWER COMMENTS

Reviewer #1 (Remarks to the Author):

Lee et al. synthesized a pair of tDPAtDIDCz isomers that contained a tDIDCz MR core and two tDPA donors and studied the photophysical and electroluminescent properties of these materials. It was found that 2, 5- and 1, 6-substitution of tDPA group in tDIDCz results in forming different degrees of CT character in excited state, which is very crucial to create efficient RISC for TADF. The 2, 5-substituted isomer exhibited a stronger CT character as supported by the photophysical and theoretical data. As a result, the emission of 2,5-tDPAtDIDCz was effectively shifted from violet region of tDIDCz to deep blue region and the singlet-triplet energy gap was dramatically reduced to 0.18 eV. The discoveries are indeed very interesting to the organic electroluminescence community. However, there are several fundamental issues that may affect the credibility of their discussion/interpretation. Therefore, I am unable to recommend the manuscript for publication in its current form.

1. The ΔE_{ST} of 2,5-tDPAtDIDCz is 0.18 eV. If the RISC does indeed happen, experimental evidence only suggested that RISC is between T1 and S1 of the MR-TADF motif. This means that Figure 2 is incorrect, particularly the level of theory required to obtain the correct ordering/energy gap of S_n and T_n resulting in incorrect interpretation/discussion of the results. The authors undertake a detailed discussion regarding the energy levels afforded by DFT calculations. Although DFT can be useful in qualitative discussions of the compounds (e.g. orbital localisations, etc), the discussion regarding singlet/triplet energy levels here is not convincing. It is known that TD-DFT can sometimes struggle to predict singlet/triplet energy levels in MR-TADF materials accurately (for an in depth discussion, see *J. Chem. Theory Comput.* 2022, 18, 8, 4903-4918). The authors should discuss their choice of functional/relative benchmarks in more detail if any quantitative conclusions are to be drawn from the reported energy levels.

2. The authors claimed that SVC mechanism might facilitate RISC via high-lying triplet state. Generally, different orbital type between singlet and triplet is the key to an efficient RISC (*J. Am. Chem. Soc.* 2017, 139, 4042). Looking at Figure 2, only a little change in orbital better S1 and T3 of 2,5-tDPAtDIDCz, which is less than that of 1,6-tDPAtDIDCz. These results are opposite to the trend of TADF efficiency of the two isomers.

3. There is no such thing called MR excited states. Short-range CT or locally excited state is a better description (*Nat. Photonics* 2021, 15, 780). So, "hybridization of the CT and MR excited states" is suggested to replace with hybridization of CT and LE excited state (HLCT) for the MR materials (*Adv. Funct. Mater.* 2023, 33, 2211893).

4. In Figure 4a, 1,6-tDPAtDIDCz showed a biexponential transient PL delay profile with a long tail beyond 200 ns. Furthermore, the use of bicomponent mCP:TSPO1 host may have an exciplex character that exists RISC channel for TADF. The guest-host interaction should be considered.

5. The authors need to explain clearer the statement: ".....Therefore, 2,5-tDPAtDIDCz may show TADF emission, while 2,5-tDPAtDIDCz may exhibit conventional fluorescence." On page 8. Also what do you mean by "disconnected conjugation structure" on page 11?

Reviewer #2 (Remarks to the Author):

The authors developed a deep-blue MR type emitter by managing the excited states of tDIDCz using a tDPA excited state manager which enables hybridized CT and MR excited state. The device exhibited a high external quantum efficiency of 30.8% and a narrow emission with a full width at half maximum of 38 nm, and a color coordinate of (0.13, 0.13). This manuscript is suggested to be accepted after addressing the following issues:

1. The characterization of material photophysical properties and OLED are brief. The CIE coordinates, EQE-Luminance and PE-Luminance curve should be presented in the text.
2. In the device performance table, the Von should also be listed.
3. Please provide the rationale of choosing the two specific materials 2,5-tDPAtDIDCz and 1,6-tDPAtDIDCz as a comparison, since 1,6-tDPAtDIDCz has much lower performance.
4. The device fabrication procedure is too brief. Please add detailed information in the device fabrication section.
5. In the introduction section, please describe the position of this work in terms of device performance compared with other similar works. Better cite ref J. Semicond., 2022, 43(11), 110202.

Reviewer #3 (Remarks to the Author):

My comments are summarized in attached document herein.

This manuscript is the extension work previously studied by Lee and co-workers reported in *Small Science* in 2020. In this contribution, the authors revealed the introduction of the tDPA units at the 1,6- or 2,5- position onto tDIDCz that could induce large discrepancy of the MOs distribution and photophysical properties. For instance, 2,5-tDPAtDIDCz was found to possess a small gap between ΔE_{ST} and ΔE_{TT} and hence more efficient direct spin-orbit coupling (long range CT) and spin-vibronic coupling assisted RISC processes (short range MR). In practice, OLED device based on the 2,5-tDPAtDIDCz demonstrated high external quantum efficiency of up to 30% and blue emission peaking at around 470 nm with FWHM of 38 nm.

In fact, the use of multiple charge-transfer excited state to induce efficient and stable thermally activated delayed fluorescence was first demonstrated by Adachi et al in 2018. A second type of electron-donating unit in a donor–acceptor system induces effective charge transfer and locally excited triplet states, resulting in an acceleration of the RISC process while maintaining high photoluminescence quantum yield (*Adv. Sci.*, **2018**, 4, eaa06910). Based on the similar approaches, TADF or MRTADF emitters bearing multiple donors for fine-tuning their excited states alignment to enhance the efficiency of the RISC process of up to $10^5 - 10^6 \text{ s}^{-1}$ has been reported recently, including but not limited the following publications: *J. Mater. Chem. C*, 2023, 11, 4210; *Angew. Chem. Int. Ed.* 2022, 61, e202209984; *Angew. Chem. Int. Ed.* 2022, 61, e202201588. On the whole, the use of tDPA as an excited state manager to the MR type tDIDCz core for manipulating energy alignment of the excited states maybe not a new concept to the OLED communality. Meanwhile, As donor materials such as carbazole, indolocarbazole or triphenylamine derivatives demonstrate the similar donating ability that could impose similar effect on this system, the authors also fail to point out the uniqueness of tDPA as an excited state manager. It is highly anticipated that the author could show the readers the reason behind the molecular design, in addition to the attachment of tDPA to different positions of molecule as illustrated in the manuscript. On the whole, I am afraid the novelty of this manuscript does not meet the standard of Nature Communication. In this regard, I have several concerns to raise as follow:

1. The excited state characters of S_1 , T_1 and T_3 should be shown in the figure 1, the relative energies of these states are clearly out of scale. Please update the figure.

2. In paragraph 14, it is not enough to explain the improved EQE besides outcoupling effect. Is there any possible evidence that prove the enhancement of EQE? For example, the PLQY in mBisPCz-O-BN thin film etc... Host polarity does affect ¹CT state of emitter and hence energy level splitting for TADF emission (*Adv. Optical Mater.* **2022**, 10, 2101343).
3. In Table 2, what is the device result of 1,6-tDPA_tDIDCz in mBisPCz-O-BN host? As suggested that SVC assisted RISC is disabled due to large difference of ΔE_{ST} and ΔE_{TT} , different host polarity could have improved such problem for efficient TADF emission by 1,6-tDPA_tDIDCz.
4. The authors mentioned that the PLQYs of 2,5-tDPA_tDIDCz and 1,6-tDPA_tDIDCz were 88 and 92%, respectively, indicating efficient radiative transition assisted by the MR core structure. What is the rationale or evidence of this statement?
5. Following the pervious question, the increased CT character in the 2,5-tDPA_tDIDCz emitter through the tDPA excited state manager did not largely degrade the PLQY. However, I found that the PLQY of tDPA is 60% in the solution state. The authors should provide the PLQY of tDPA in 1,3-di(9*H*-carbazol-9-yl)benzene (mCP) : diphenyl(4-(triphenylsilyl)phenyl)phosphine oxide (TSPO1).
6. In the experimental section of device fabrication: PEDOT:PSS layer in device is produced by wet coating technique. Please specify such fabrication process as well.
7. The operational stability result of the OLED should be provided.
8. Some typing problems are found in the manuscripts, for examples, "Therefore, 2,5-tDPA_tDIDCz may show TADF emission, while 2,5-tDPA_tDIDCz may exhibit conventional fluorescence." Please check thoroughly.

Dear reviewers

We would like to express our gratitude to the reviewers who spent valuable time to evaluate our work titled “Hybridized Multiple Resonance and Charge Transfer Excited States Using an Excited State Manager for Efficient Reverse Intersystem Crossing and Narrow Emission”. We have revised the manuscript according to your helpful comments. I have attached the revised manuscript along with the response letter.

Reviewer #1

Lee et al. synthesized a pair of tDPAtDIDCz isomers that contained a tDIDCz MR core and two tDPA donors and studied the photophysical and electroluminescent properties of these materials. It was found that 2, 5- and 1, 6-substitution of tDPA group in tDIDCz results in forming different degrees of CT character in excited state, which is very crucial to create efficient RISC for TADF. The 2, 5-substituted isomer exhibited a stronger CT character as supported by the photophysical and theoretical data. As a result, the emission of 2,5-tDPAtDIDCz was effectively shifted from violet region of tDIDCz to deep blue region and the singlet-triplet energy gap was dramatically reduced to 0.18 eV. The discoveries are indeed very interesting to the organic electroluminescence community. However, there are several fundamental issues that may affect the credibility of their discussion/interpretation. Therefore, I am unable to recommend the manuscript for publication in its current form.

1. The ΔE_{ST} of 2,5-tDPAtDIDCz is 0.18 eV. If the RISC does indeed happen, experimental evidence only suggested that RISC is between T1 and S1 of the MR-TADF motif. This means that Figure 2 is incorrect, particularly the level of theory required to obtain the correct ordering/energy gap of S_n and T_n resulting incorrect interpretation/discussion of the results. The authors undertake a detailed discussion regarding the energy levels afforded by DFT calculations. Although DFT can be useful in qualitative discussions of the compounds (e.g. orbital localisations, etc), the discussion regarding singlet/triplet energy levels here is not convincing. It is known that TD-DFT can sometimes struggle to predict singlet/triplet energy levels in MR-TADF materials accurately (for an in depth discussion, see **J. Chem. Theory Comput.** **2022**, **18**, **8**, **4903-4918**). The authors should discuss their choice of functional/relative benchmarks in more detail if any quantitative conclusions are to be drawn from the reported energy levels.

Response

I appreciate your comments and understand your concerns. According to your comments, we conducted in-depth discussion and analysis of the issues regarding the computational analysis. It is meaningful to consider wave function method (e.g. SCS-CC2, EOM-CCSD) based on the paper (*J. Chem. Theory Comput.* 2022, 18, 4903–4918). This claim is best illustrated by the spin density plots in Figure 3 of the revised paper. When an electron donor moiety is introduced into the MR core (e.g. DABNA) functioning as an electron acceptor, the TD(A)-B3LYP level analyzes the S_1 excited state of MR-TADF by long-range charge transfer (LRCT). However, the actual S_1 excited state of MR-TADF materials usually exhibits short-range charge transfer (SRCT). Through these examples, we believe that the methodology based on the wave function method shows high reliability for calculating boron type MR-TADF emitters.

Therefore, to verify the accuracy of the single point calculation performed at the MPM1B95/G(d,p) level in our manuscript, we additionally performed *ab initio* calculation using various methods: TD-DFT calculation at B3LYP, M06-2X, gap-tuned ω B97XD*/g(d,p) level, and wave function method performed at EOM-CCSD/cc-pVDZ level. The errors between the calculated excitation energies and the experimental values (singlet energy; E_S , triplet energy; E_T , singlet-triplet energy splitting; ΔE_{ST}) are summarized in the table below.

Structure	Single Point Energy Calculation @ Optimized S ₀ Geometry at B3LYP/G(d,p) level										: Smallest error				Error = Exp - Cal	
	Functional	Basis	S ₁ (eV)	T ₁ (eV)	T ₂ (eV)	T ₃ (eV)	T ₄ (eV)	T ₅ (eV)	ΔE _{ST} (eV)	E _S (Exp) (eV)	Error (eV)	E _T (Exp) (eV)	Error (eV)	ΔE _{ST} (Exp) (eV)	Error (eV)	
 2,5-tDPAtdIDCz	B3LYP	6-31G(d,p)	2.757	2.458	2.647	2.744	2.861	3.031	0.299	2.83	0.073	2.65	0.192	0.18	-0.119	
	M06-2X	6-31G(d,p)	3.402	2.973	3.183	3.294	3.372	3.491	0.428		-0.572		-0.323			
	ω B97XD ^a	6-31G(d,p)	3.134	2.720	2.852	2.914	3.067	3.186	0.414		-0.304		-0.070			
	EOM-CCSD	cc-pVDZ	4.300	2.598	2.827	2.972	3.043	3.142	1.702		-1.470		0.052			
	MPW1B95	6-31G(d,p)	2.761	2.467	2.651	2.755	2.869	2.986	0.294		0.069		0.183			
 1,6-tDPAtdIDCz	B3LYP	6-31G(d,p)	2.947	2.399	2.782	2.948	3.023	3.063	0.548	2.99	0.043	2.58	0.181	0.41	-0.138	
	M06-2X	6-31G(d,p)	3.497	2.846	3.257	3.366	3.489	3.528	0.651		-0.507		-0.266			
	ω B97XD ^b	6-31G(d,p)	3.260	2.553	2.905	3.104	3.192	3.252	0.707		-0.270		0.027			
	EOM-CCSD	cc-pVDZ	4.427	2.732	2.883	3.083	3.167	3.173	1.695		-1.437		-0.152			
	MPW1B95	6-31G(d,p)	2.950	2.395	2.782	2.940	3.017	3.064	0.555		0.040		0.185			

^a Range separation parameter, ω value was optimally tuned as 0.1052. ^b Range separation parameter, ω value was optimally tuned as 0.1030. Both parameters were obtained by following method: *J. Chem. Theory Comput.* 2015, 11, 3851–3858.

A comparison between methods is as follows. First, the EOM-CCSD/cc-pVDZ level provided a modest error in estimating E_T , but an abnormally large E_S . For this reason, we excluded the results of the EOM-CCSD/cc-pVDZ level from the comparison between *ab initio* calculation methods. Overall, the B3LYP and MPW1B95 levels provided similar and significantly smaller errors in E_S , E_T and ΔE_{ST} . The gap-tuned ω B97XD* level showed high similarity with the experimental value for E_T . However, it predicted large E_S , which seems to be due to the underestimated long-range CT interaction. The M06-2X level provided a similar level of ΔE_{ST} to the gap-tuned ω B97XD* level, but both E_S and E_T showed large error. In summary, the results of this series of calculations provide a conclusion that the TD-DFT calculation at MPW1B95/G(d,p) level that we provided in the manuscript is not wrong, but rather provides a higher level of accuracy than other methods.

2. The authors claimed that SVC mechanism might facilitate RISC via high-lying triplet state. Generally, different orbital type between singlet and triplet is the key to an efficient RISC (J. Am. Chem. Soc. 2017, 139, 4042). Looking at Figure 2, only a little change in orbital better S1 and T3 of 2,5-tDPA_tDIDCz, which is less than that of 1,6-tDPA_tDIDCz. These results are opposite to the trend of TADF efficiency of the two isomers.

Response

I appreciate your comments. The paper provided by the reviewer demonstrates the importance of the origin of singlet and triplet excited states to induce efficient RISC of TADF materials (specifically, D-A type TADF). In the D-A type emitters, the T_1 state is required to possess locally excited (LE) character for efficient RISC by El-sayed rule because D-A type TADF materials clearly possess charge transfer (CT) character in S_1 state. In line with this, the natural transition orbital (NTO) distributions of S_1 and T_3 states are consistent with the proposal of the previous paper, rationalizing the SVC-RISC mechanism.

2,5-tDPAAtDIDCz

1,6-tDPAAtDIDCz

The above figure shows the NTO distributions at S_1 and T_3 states that we provided in **Figure 2**, **Figure S2** and **Figure S3** (blue occupied: HONTO, red occupied: LUNTO). LUNTO is mainly distributed in DIDCz core that functions as an electron acceptor, while tDPA units contribute to the formation of HONTO. However, in the case of 1,6-tDPAAtDIDCz, the DIDCz core also contributes to the HONTO, providing significant HONTO-LUNTO overlap. On the other hand, in the case of 2,5-tDPAAtDIDCz, the contribution of DIDCz core to HONTO formation is negligible. From this, we can see that the S_1 state of 2,5-tDPAAtDIDCz can be regarded as CT excited state, and the S_1 state of 1,6-tDPAAtDIDCz can be regarded as a hybridized LE-CT excited state. In fact, photoluminescence and EL spectra revealed that a LE featured emission with a narrow FWHM is observed in 1,6-tDPAAtDIDCz, and a relatively broad CT featured emission is observed in 2,5-tDPAAtDIDCz (**Figure 3** and **Figure 5 (b)**).

It is advantageous that the origin of excited state of the T_3 state, which is the high lying T_n state that contributes to SVC-RISC, is the LE state judging from the localized HONTO and LUNTO on the DIDCz core. Therefore, the T_3 state of the two emitters can mediate SVC-RISC. However, 1,6-tDPAAtDIDCz did not exhibit TADF characteristic due to large ΔE_{ST} and ΔE_{TT} .

In conclusion, 2,5-tDPA_tDIDCz and 1,6-tDPA_tDIDCz have electronic structures consistent with the TADF behavior of them.

Added sentence (1)

In addition, the T₃ state of 2,5-tDPA_tDIDCz had an excitation energy similar to that of the S₁ state and it is predicted to be a LE dominant state because HONTO and LUNTO are distributed in DIDCz core. This implies that the close-lying T₃ state is suitable as the SVC-mediating high-lying T_n state for CT dominant S₁ state to induce efficient RISC.

Added sentence (2)

In summary, 2,5-tDPA_tDIDCz exhibited sufficiently small ΔE_{ST} due to the introduction of the excited state manager, and the reconstructed triplet excited states provided appropriate NTO distributions and SOCME values suitable for harvesting triplet exciton *via* direct SOC and 2nd-order RISC. On the other hand, as can be seen in 1,6-tDPA_tDIDCz, the linearly extended arrangement of two tDPA units could not build proper electronic structure suitable for triplet exciton harvesting.

3. There is no such thing called MR excited states. Short-range CT or locally excited state is a better description (Nat. Photonics 2021, 15, 780). So, “hybridization of the CT and MR excited states” is suggested to replace with hybridization of CT and LE excited state (HLCT) for the MR materials (Adv. Funct. Mater. 2023, 33, 2211893).

Response

I appreciate your comment. According to your comment, we changed article title and MR excited state to short range CT state.

Modified title

Hybridization of Short-Range and Long-Range Charge Transfer Excited States on Multiple Resonance Emitter Using an Excited State Manager for Efficient Reverse

Intersystem Crossing and Narrow Emission

Modified sentence

The excited state manager red-shifted the emission spectrum from violet to deep blue region and opened the RISC channel for TADF emission by hybridizing long-range CT (LRCT) and short-range CT (SRCT) excited state character.

It was demonstrated that direct RISC by LRCT and spin-vibronic coupling (SVC) assisted RISC by SRCT through MR core structure simultaneously contributed to the up-conversion of triplet excitons.

This work proposed that the hybridization of LRCT and SRCT excited states can realize both high EQE and narrow emission spectrum.

The hybridization concept of LRCT and SRCT excited states is to utilize the narrow emission from MR character and efficient RISC from CT character as presented in **Figure 1**.

Therefore, the S_1 energy of 2,5-tDPAtDIDCz was significantly lowered by LRCT interaction between *para*-oriented sp^2 nitrogen of ICz and sp^3 nitrogen of tDPA.

The slightly broadened spectrum and weak shoulder in the 2,5-tDPAtDIDCz propose combined emissions from LRCT and SRCT excited states.

The participation of the rigid MR chromophore for fluorescence sharpened the emission spectrum while achieving high EQE by TADF through combined CT properties.

In this work, deep-blue MR type emitters were developed by managing the excited states of tDIDCz using a tDPA excited state manager which enables hybridized LRCT and SRCT excited state.

As a result of hybridization of **LRCT and SRCT** excited state,

4. In Figure 4a, 1,6-tDPAtDIDCz showed a biexponential transient PL delay profile with a long tail beyond 200 ns. Furthermore, the use of bicomponent mCP:TSPO1 host may have an exciplex character that exists RISC channel for TADF. The guest-host interaction should be considered.

Response

I appreciate your comment. In fact, mCP:TSPO1 is not an exciplex host. The results of the previously reported paper (*Mater. Horiz.*, 2022, **9**, 1299–1308) support this (please refer to **Figure S10**, solid PL spectra). Therefore, it is considered unlikely that mCP:TSPO1 host system activates additional RISC channel.

We reported that 1,6-tDPAtDIDCz is a deep blue fluorescent emitter with no TADF characteristics due to large ΔE_{ST} . To prove this, the exciton decay curve was recorded up to 1 ms time range. Consequently, no TADF characteristic was observed for this material.

Added supporting figure

Figure S6. Delayed component decay curve of 1,6-tDPAtDIDCz doped in mCP:TSPO1 host.

Added and modified sentence

The tDIDCz emitter was a conventional fluorescence emitter without any TADF emission, but it was transformed into a TADF emitter by modifying 2 and 5-positions of tDIDCz with tDPA, whereas 1 and 6-position modification of tDIDCz with tDPA did not deliver TADF characteristic (**Figure S6**).

5. The authors need to explain clearer the statement: ".....Therefore, 2,5-tDPA_tDIDCz may show TADF emission, while 2,5-tDPA_tDIDCz may exhibit conventional fluorescence." On page 8. Also what do you mean by “disconnected conjugation structure” on page 11?

Response

I appreciate your comment. We corrected the mistake. The corresponding sentence explained that the triplet energy of 2,5-tDPA_tDIDCz was larger than that of 1,6-tDPA_tDIDCz in terms of molecular structure. 1,6-tDPA_tDIDCz has a chemical structure with two tDPA units aligned through para position of the aromatic unit in a linear fashion. Therefore, the conjugation is extended. In the case of 2,5-tDPA_tDIDCz, the two tDPA units are attached to the meta position of the through the aromatic units in a bent fashion. Therefore, the conjugation extension is disrupted.

Modified sentence

Therefore, 2,5-tDPA_tDIDCz may show TADF emission, while 1,6-tDPA_tDIDCz may exhibit conventional fluorescence.

Reviewer #2

The authors developed a deep-blue MR type emitter by managing the excited states of tDIDCz using a tDPA excited state manager which enables hybridized CT and MR excited state. The device exhibited a high external quantum efficiency of 30.8% and a narrow emission with a full width at half maximum of 38 nm, and a color coordinate of (0.13, 0.13). This manuscript is suggested to be accepted after addressing the following issues:

1. The characterization of material photophysical properties and OLED are brief. The CIE coordinates, EQE-Luminance and PE-Luminance curve should be presented in the text.

Response

I appreciate your comment. We added EQE-L and PE-L curves to specify the device characteristics. In addition, the device data of the newly fabricated mBisPCz-O-BN:1,6-tDPA:tDIDCz device was also added. Table 2 was modified to specify their device characteristics, CIE color coordinates and turn-on voltage (for response of comment 2). The added EQE-L and PE-L curves are illustrated in **Figure S10** and **Figure S11**.

Modified table

Table 2. Summarized device performance of the blue OLEDs.

Compound		V_{on}^a (V)	V_d^b (V)	λ_{EL}^b (nm)	FWHM ^b (nm)	CIE (x, y) ^b	EQE ^c (%)	PE ^c (lm W ⁻¹)	CE ^c (cd A ⁻¹)
2,5-tDPA:tDIDCz	1% ^d	3.7	4.7	464	36	(0.13, 0.12)	23.4	23.1	24.2
	1% ^e	3.5	4.8	466	38	(0.13, 0.14)	30.8	31.5	34.1
1,6-tDPA:tDIDCz	1% ^d	4.1	5.4	437	32	(0.16, 0.04)	6.4	3.1	3.4
	1% ^e	3.9	6.1	438	36	(0.16, 0.05)	7.3	8.8	8.7

^a Turn-on voltage at a luminance of 1 cd m⁻². ^b Driving voltage at a luminance of 100 cd m⁻². ^c Maximum values.

^d 1 wt% doped mCP:TSPO1 hosted device. ^e 1 wt% doped mBisPCz-O-BN hosted device.

Added supporting figures

Figure S10. The device performance of mCP:TSP01 hosted devices. (a) External quantum efficiency-luminance curves and (b) power efficiency-luminance curves of 2,5-tDPAAtDIDCz (blue) and 1,6-tDPAAtDIDCz (black).

Figure S11. The device performance of mBisPCz-O-BN hosted devices. (a) External quantum efficiency-luminance curves and (b) power efficiency-luminance curves of 2,5-tDPAAtDIDCz (blue) and 1,6-tDPAAtDIDCz (black).

Added sentence

The EQE-luminance and power efficiency-luminance curves of mCP:TSP01 hosted devices and mBisPCz-O-BN hosted devices are illustrated in **Figure S10** and **Figure S11**.

2. In the device performance table, the V_{on} should also be listed.

Response

I appreciate your comment. We added turn-on voltage (V_{on}) and driving voltage (V_d) of fabricated devices in modified Table 2. Please check the response of comment 1.

3. Please provide the rationale of choosing the two specific materials 2,5-tDPA-tDIDCz and 1,6-tDPA-tDIDCz as a comparison, since 1,6-tDPA-tDIDCz has much lower performance.

Response

I appreciate your comment. We developed two isomeric blue emitters, 2,5-tDPA-tDIDCz and 1,6-tDPA-tDIDCz, by introducing the tDPA unit into the MR core. Our motivation was to systematically study the effect of the donor substitution position on the emission properties of the MR emitters. Despite similar chemical structure, large differences of photophysical properties and device performances were observed.

Added sentence

Different effects of donor substitution position (2,5- or 1,6- position) on tDIDCz core provided formation of significantly different electronic structures.

4. The device fabrication procedure is too brief. Please add detailed information in the device fabrication section.

Response

I appreciate your comment. We added the information about detailed device fabrication procedure.

Added and modified sentences

Device Fabrication

The bottom emission device structure of blue OLED was indium tin oxide (ITO) (50 nm) / PEDOT:PSS (40 nm) / TAPC (5 nm) / TCTA (5 nm) / PCzAc (5 nm) / mCP (5 nm) / mCP:TSPO1:2,5-tDPA:tDIDCz or 1,6-tDPA:tDIDCz (25 nm, 50%, 1 wt%) / TSPO1 (25 nm) / LiF (1.5 nm)/Al (200 nm). The optimized device structure was ITO (50 nm) / PEDOT:PSS (40 nm) / TAPC (10 nm) / mCP (10 nm) / mBisPCz-O-BN:2,5-tDPA:tDIDCz (25 nm, 1 wt%) / TSPO1 (25 nm) / LiF (1.5 nm)/Al (200 nm). The molecular structure of organic electronic materials used in blue OLED devices are illustrated in **Figure S9**. The ITO substrates were cleaned by ultrasonication with acetone, isopropylalcohol, chloroform and deionized water and dried in oven. Also, oxygen plasma treatment to ITO substrates was performed to fine-tune the work function of anode. A 40 nm thick PEDOT:PSS film was formed on ITO substrate by spin-coating (30 s at 3,200 rpm) followed by annealing at 150 °C for 15 min). Vacuum thermal deposition process was proceeded under a pressure of 2.0×10^{-7} Torr and the devices were encapsulated with a glass lid in nitrogen-filled glove box. Device characteristics were recorded by using a CS2000 spectroradiometer and Keithley 2400 for electrical and optical output measurement. PEDOT:PSS stands for poly(3,4-ethylenedioxythiophene)-poly(styrenesulfonate). TAPC is (1,1-bis(4-di-*p*-tolylaminophenyl)cyclohexane), TCTA is tris(4-carbazoyl-9-ylphenyl)amine, and PCzAc is 9,9-dimethyl-10-(9-phenyl-9*H*-carbazol-3-yl)-9,10-dihydroacridine.

5. In the introduction section, please describe the position of this work in terms of device performance compared with other similar works. Better cite ref J. Semicond., 2022, 43(11), 110202.

Response

I appreciate your comment. We added summarized device performance data reported in reported papers. A table was added to the supporting information as **Table S3**. Reported device

data of D-A type TADF, MR-TADF and ICz derivatives with CIE_y coordinate less than 0.20 were collected.

Added supporting table

Table S3. Summarized device performances of reported blue OLEDs.

Emitter	Emitter type	λ_{EL}^a (nm)	$FWHM_{EL}^b$ (nm)	EQE_{Max}^c (%)	CIE (x, y) ^d	Ref
CzBPCN		460	48	14	(0.14, 0.12)	S2
TDBA-Ac		445	48	21.5	(0.15, 0.06)	
TDBA-Ac	DA-TADF ^e	464	55	25.7	(0.14, 0.15)	S3
DMAC2PTO		465	56	32.2	(0.14, 0.15)	
TDBA-SAF		448	52	15.2	(0.154, 0.108)	S4
TDBA-PAS		456	55	28.2	(0.142, 0.090)	S5
TDBA-DPAC		435	50	22.35	(0.155, 0.042)	S6
TDBA-DPAC		449	59	21.32	(0.150, 0.077)	
DABNA-1		459	28	13.5	(0.13, 0.09)	18
DABNA-TP-TB		457	33	19.5	(0.14, 0.11)	S7
v-DABNA		469	18	34.4	(0.12, 0.11)	19
V-DABNA-F8		468	15	26.6	(0.09, 0.10)	20
v-DABNA-O-Me		465	23	29.5	(0.13, 0.10)	S8
2B-DTACrs		447	26	14.8	(0.150, 0.044)	S9
m-v-DABNA		471	18	36.2	(0.12, 0.12)	
4F-v-DABNA		464	18	35.8	(0.13, 0.10)	S10
4F-m-v-DABNA		461	18	33.7	(0.13, 0.06)	
BOBO-Z		445	18	13.6	(0.15, 0.04)	
BOBS-Z	MR-TADF ^f	456	23	26.9	(0.14, 0.06)	S11
BSBS-Z		463	22	26.8	(0.13, 0.08)	
t-DAB-DPA		459	26	27.9	(0.13, 0.08)	S12
3tPAB		460	26	19.3	(0.14, 0.08)	S13
m-DINBO		466	21	24.2	(0.126, 0.098)	S14
BBCz-DB		469	27	29.3	(0.12, 0.18)	S15
QA-1		455	39	17.1	(0.14, 0.12)	S16
CzBO		448	30	13.4	(0.15, 0.05)	
CzBS		473	31	23.1	(0.11, 0.16)	S17
tCBNDADPO		468	24	13.8	(0.12, 0.16)	S18
mBP-DABNA-Me		468	28	24.3	(0.12, 0.14)	S19
R-DOBN		464	35	25.6	(0.13, 0.12)	S20
tDIDCz		401	14	2.75	(0.164, 0.018)	21
BisICz		437	24	6.5	(0.16, 0.04)	
tBisICz		445	22	15.1	(0.16, 0.05)	15
tPBisICz		452	21	23.1	(0.15, 0.05)	
t3IDCz	ICz derivatives ^g	472	25	30.0	(0.119, 0.161)	
p3IDCz		472	23	30.9	(0.120, 0.158)	30
2,5-tDPA tDIDCz		464	36	23.4	(0.13, 0.12)	
		466	38	30.8	(0.13, 0.14)	this work
1,6-tDPA tDIDCz		437	32	6.4	(0.16, 0.04)	this work
		438	36	7.3	(0.16, 0.05)	

^a Peak wavelength of electroluminescence (EL). ^b Full-width-at-half-maximum of EL spectrum. ^c Maximum external quantum efficiency. ^d CIE color coordinates. ^e Donor-acceptor type TADF. ^f Multiple-resonance type TADF. ^g Indolo[3,2,1-*jk*]carbazole derived MR type emitters.

Added supporting references

- S2. Cho YJ, Jeon SK, Lee S-S, Yu E, Lee JY. Donor interlocked molecular design for fluorescence-like narrow emission in deep blue thermally activated delayed fluorescent emitters. *Chem. Mater.* **28**, 5400-5405 (2016).
- S3. Ahn DH, et al. Highly efficient blue thermally activated delayed fluorescence emitters based on symmetrical and rigid oxygen-bridged boron acceptors. *Nat. Photon.* **13**, 540-546 (2019).
- S4. Sun S, et al. Efficient deep-blue thermally activated delayed fluorescence emitters based on diphenylsulfone-derivative acceptor. *Dyes Pigm.* **178**, 108367 (2020).
- S5. Lim H, Cheon HJ, Woo SJ, Kwon SK, Kim YH, Kim JJ. Highly Efficient Deep-Blue OLEDs using a TADF Emitter with a Narrow Emission Spectrum and High Horizontal Emitting Dipole Ratio. *Adv. Mater.* **32**, 2004083 (2020).
- S6. Tan HJ, et al. Deep-Blue OLEDs with Rec. 2020 Blue Gamut Compliance and EQE Over 22% Achieved by Conformation Engineering. *Adv. Mater.* **34**, 2200537 (2022).
- S7. Oda S, Kumano W, Hama T, Kawasumi R, Yoshiura K, Hatakeyama T. Carbazole-Based DABNA Analogues as Highly Efficient Thermally Activated Delayed Fluorescence Materials for Narrowband Organic Light-Emitting Diodes. *Angew. Chem.* **133**, 2918-2922 (2021).
- S8. Tanaka H, et al. Hypsochromic shift of multiple-resonance-induced thermally activated delayed fluorescence by oxygen atom incorporation. *Angew. Chem. Int. Ed.* **60**, 17910-17914 (2021).
- S9. Chan C-Y, et al. Two boron atoms versus one: high-performance deep-blue multi-resonance thermally activated delayed fluorescence emitters. *Chem. Comm.* **58**, 9377-9380 (2022).
- S10. Naveen KR, Lee H, Braveenth R, Yang KJ, Hwang SJ, Kwon JH. Deep blue diboron embedded multi-resonance thermally activated delayed fluorescence emitters for narrowband organic light emitting diodes. *Chem. Eng. J.* **432**, 134381 (2022).
- S11. Park IS, Yang M, Shibata H, Amanokura N, Yasuda T. Achieving Ultimate Narrowband and Ultrapure Blue Organic Light-Emitting Diodes Based on Polycyclo-Heteraborin Multi-Resonance Delayed-Fluorescence Emitters. *Adv. Mater.* **34**, 2107951 (2022).
- S12. Kim J, Chung W, Kim J, Lee J. Concentration quenching-resistant multiresonance thermally activated delayed fluorescence emitters. *Mater. Today Energy* **21**, 100792 (2021).
- S13. Wang Y, et al. A periphery cladding strategy to improve the performance of narrowband emitters, achieving deep-blue OLEDs with CIEy < 0.08 and external quantum efficiency approaching 20%. *Org. Electron.* **97**, 106275 (2021).
- S14. Liu G, Sasabe H, Kumada K, Arai H, Kido J. Nonbonding/Bonding Molecular Orbital Regulation of Nitrogen-Boron-Oxygen-embedded Blue/Green Multiresonant TADF Emitters with High Efficiency and Color Purity. *Chem. Eur. J.* **28**, e202201605 (2022).
- S15. Yang M, Park IS, Yasuda T. Full-color, narrowband, and high-efficiency electroluminescence from boron and carbazole embedded polycyclic heteroaromatics.

J. Am. Chem. Soc. **142**, 19468-19472 (2020).

- S16. Min H, Park IS, Yasuda T. cis-Quinacridone-Based Delayed Fluorescence Emitters: Seemingly Old but Renewed Functional Luminogens. *Angew. Chem.* **133**, 7721-7726 (2021).
- S17. Park IS, Min H, Yasuda T. Ultrafast Triplet–Singlet Exciton Interconversion in Narrowband Blue Organoboron Emitters Doped with Heavy Chalcogens. *Angew. Chem.* **134**, e202205684 (2022).
- S18. Bian J, et al. Ambipolar Self-Host Functionalization Accelerates Blue Multi-Resonance Thermally Activated Delayed Fluorescence with Internal Quantum Efficiency of 100%. *Adv. Mater.* **34**, 2110547 (2022).
- S19. Cheon HJ, Shin YS, Park NH, Lee JH, Kim YH. Boron-Based Multi-Resonance TADF Emitter with Suppressed Intermolecular Interaction and Isomer Formation for Efficient Pure Blue OLEDs. *Small* **18**, 2107574 (2022).
- S20. Yan ZP, et al. A Chiral Dual-Core Organoboron Structure Realizes Dual-Channel Enhanced Ultrapure Blue Emission and Highly Efficient Circularly Polarized Electroluminescence. *Adv. Mater.* **34**, 2204253 (2022).

Added sentence

Table 2 provides all device performances reported in this work, and **Table S3** provides the summarized device performance of the reported blue OLEDs.

Reviewer #3

This manuscript is the extension work previously studied by Lee and co-workers reported in Small Science in 2020. In this contribution, the authors revealed the introduction of the tDPA units at the 1,6- or 2,5- position onto tDIDCz that could induce large discrepancy of the MOs distribution and photophysical properties. For instance, 2,5-tDPAtDIDCz was found to possess a small gap between Δ_{EST} and Δ_{ETT} and hence more efficient direct spin-orbit coupling (long range CT) and spin-vibronic coupling assisted RISC processes (short range MR). In practice, OLED device based on the 2,5-tDPAtDIDCz demonstrated high external quantum efficiency of up to 30% and blue emission peaking at around 470 nm with FWHM of 38 nm. In fact, the use of multiple charge-transfer excited state to induce efficient and stable thermally activated delayed fluorescence was first demonstrated by Adachi et al in 2018. A second type of electron-donating unit in a donor-acceptor system induces effective charge transfer and locally excited triplet states, resulting in an acceleration of the RISC process while maintaining high photoluminescence quantum yield (Adv. Sci., 2018, 4, eaao6910). Based on the similar approaches, TADF or MRTADF emitters bearing multiple donors for fine-tuning their excited states alignment to enhance the efficiency of the RISC process of up to $10^5 - 10^6 \text{ s}^{-1}$ has been reported recently, including but not limited the following publications: J. Mater. Chem. C, 2023, 11, 4210; Angew. Chem. Int. Ed. 2022, 61, e202209984; Angew. Chem. Int. Ed. 2022, 61, e202201588. On the whole, the use of tDPA as an excited state manager to the MR type tDIDCz core for manipulating energy alignment of the excited states maybe not a new concept to the OLED community. Meanwhile, As donor materials such as carbazole, indolocarbazole or triphenylamine derivatives demonstrate the similar donating ability that could impose similar effect on this system, the authors also fail to point out the uniqueness of tDPA as an excited state manager. It is highly anticipated that the author could show the readers the reason behind the molecular design, in addition to the attachment of tDPA to different positions of molecule as illustrated in the manuscript. On the whole, I am afraid the novelty of this manuscript does not meet the standard of Nature Communication. In this regard, I have several concerns to raise as follow:

1. The excited state characters of S1, T1 and T3 should be shown in the figure 1, the relative energies of these states are clearly out of scale. Please update the figure.

Response

I appreciate your comment. We modified figure 1 to clearly specify the exact excited state energy. The excitation energy was referred to calculated TD-DFT simulation data conducted at B3LYP/G(d) level.

Single Point Calculation @ B3LYP/G(d) level							
Name	S ₁ (eV)	T ₁ (eV)	T ₂ (eV)	T ₃ (eV)	T ₄ (eV)	T ₅ (eV)	ΔE _{ST} (eV)
tDIDCz	3.3591	2.7736	2.9186	3.0185	3.0910	3.1258	0.5855
2,5-tDPA/tDIDCz	2.7643	2.4634	2.6517	2.7483	2.8656	3.0357	0.3009

Modified figure

Figure 1. Design concept of two MR emitters. The blue and red colored parts are HOMO and LUMO distribution of corresponding molecules, respectively.

2. In paragraph 14, it is not enough to explain the improved EQE besides outcoupling effect. Is there any possible evidence that prove the enhancement of EQE? For example, the PLQY in mBisPCz-O-BN thin film etc... Host polarity does affect 1CT state of emitter and hence energy level splitting for TADF emission (Adv. Optical Mater. 2022, 10, 2101343).

Response

I appreciate your comment. As the reviewer pointed out, host is known to significantly affect the RISC kinetics of TADF materials. These effects include fine-tuning the energy level of ¹CT₁ emissive state of TADF emitters and its effect on vibronic coupling behavior such as molecular rotation. To verify these effects and analyze the causes of the EQE enhancement, we prepared

mBisPCz-O-BN hosted films and further conducted PLQY and transient PL analyses. In conclusion, the mBisPCz-O-BN host system improved the PLQY of 2,5-tDPA_tDIDCz. The improved PLQY and outcoupling efficiency are responsible for the improved EQE.

Modified table in Supporting information

Table S1. Summary of photophysical properties related to radiative transition in solid matrix.

	λ_{em} (nm)	E_s (eV)	FWHM (nm)	PLQY (%)	τ_{PF} (ns)	k_{PF} ($10^7 s^{-1}$)	τ_{DF} (μs)	k_{DF} ($10^3 s^{-1}$)	Φ_{PF} (%)	Φ_{DF} (%)	k_{ISC} ($10^7 s^{-1}$)	k_{RISC} ($10^4 s^{-1}$)
2,5-tDPA _t DIDCz ^a	461	2.69	40	88	9.0	11.11	251.4	3.98	26	62	8.22	1.28
2,5-tDPA _t DIDCz ^b	464	2.67	42	92	11.6	8.62	240.5	4.16	29	63	6.12	1.27
1,6-tDPA _t DIDCz ^a	437	2.84	40	92	3.0	33.33	N.A. ^c	N.A. ^c	92	N.A. ^c	N.A. ^c	N.A. ^c

^a Measured at mCP:TSPO1 blended film with 1 wt% doping concentration under N₂ atmosphere. ^b Measured in mBisPCz-O-BN blended film with 1 wt% doping concentration under N₂ atmosphere. ^c N.A.: Not assigned.

Added figure in Supporting information

Figure S10. (a) Prompt component decay and (b) delayed component decay curves of 2,5-tDPA_tDIDCz doped in mBisPCz-O-BN host.

Added sentence

To understand the origin of improved EQE in the mBisPCz-O-BN hosted device, transient PL and PLQY analysis of the emitter doped mBisPCz-O-BN films was conducted (**Figure S10**). The PL analysis revealed that mBisPCz-O-BN:2,5-tDPA_tDIDCz film provided an improved PLQY of 92% and significantly lowered k_{ISC} while maintaining k_{RISC} . Therefore, it can be concluded that the mBisPCz-O-BN host dramatically improved the EQE of the 2,5-

tDPAtDIDCz device by enhancing the outcoupling efficiency and PLQY compared to the mCP:TSPO1 host.

3. In Table 2, what is the device result of 1,6-tDPAtDIDCz in mBisPCz-O-BN host? As suggested that SVC assisted RISC is disabled due to large difference of ΔEST and ΔETT , different host polarity could have improved such problem for efficient TADF emission by 1,6-tDPAtDIDCz.

Response

I appreciate your comment. We fabricated and evaluated a mBisPCz-O-BN hosted 1,6-tDPAtDIDCz device. The device characteristics of this newly fabricated device are provided in **Figure S8**. The recorded EQE_{Max} was 7.3%, and there was no significant difference compared to that of the mCP:TSPO1 hosted device. In other words, 1,6-tDPAtDIDCz did not activate the SVC assisted RISC process even in the mBisPCz-O-BN host, and as a result, it showed only fluorescent emission without TADF.

Modified supporting figure

Figure S8. (a) The device structure and energy level diagram of mBisPCz-O-BN hosted blue OLEDs. (b) Normalized EL spectra, (c) current density-voltage-luminance curves, (d) external quantum efficiency-current density curves of 2,5-tDPAtdIDCz (blue) and 1,6-tDPAtdIDCz (black) devices.

Added sentence

However, the mBisPCz-O-BN hosted 1,6-tDPAtdIDCz device recorded EQE_{Max} of 7.3%. Although the EQE was slightly improved, there was no significant difference compared to that of the mCP:TSPO1 hosted device. The host matrix did not activate the SVC-RISC process of 1,6-tDPAtdIDCz, and 1,6-tDPAtdIDCz device still showed only fluorescence without TADF process.

4. The authors mentioned that the PLQYs of 2,5-tDPAtdIDCz and 1,6-tDPAtdIDCz were 88 and 92%, respectively, indicating efficient radiative transition assisted by the MR core structure. What is the rationale or evidence of this statement?

Response

I appreciate your comment. It was reported that the backbone structure of 2,5-tDPA:tDIDCz and 1,6-tDPA:tDIDCz, tDIDCz, exhibited PLQY of 60% (Small 2020, 16, 1907569). In general, it is known that when an auxochromophore such as diphenylamine is introduced into the chromophore, it provides an additional $n-\pi^*$ transition to the chromophore, allowing a more efficient radiative transition. In this work, the PLQY was enhanced from 60% to about 90%.

5. Following the previous question, the increased CT character in the 2,5-tDPA:tDIDCz emitter through the tDPA excited state manager did not largely degrade the PLQY. However, I found that the PLQY of tDPA is 60% in the solution state. The authors should provide the PLQY of tDPA in 1,3-di(9H-carbazol-9-yl)benzene (mCP) : diphenyl(4-(triphenylsilyl)phenyl)phosphine oxide (TSPO1).

Response

I appreciate your comment. I cannot clearly understand the reviewer comment, but it seems that the reviewer points out how 2,5-tDPA:tDIDCz can exhibit a high PLQY of 88%, even though tDPA, a fragment of 2,5-tDPA:tDIDCz, exhibits a PLQY of 60%. The tDPA unit is just an auxochromophore that supports the MR chromophore, tDIDCz. This means that the origin of emission of 2,5-tDPA:tDIDCz is not the tDPA unit. Please refer the following solution PL emission spectrum of diphenylamine (DPhA) reported in *Talanta* 121 (2014) 239–246. Through this figure, we can confirm that the emission wavelength of diphenylamine is less than 400 nm and possess a large singlet energy. This fact suggests that even if a diphenylamine derivative such as tDPA is incorporated into the tDIDCz core, tDPA cannot be the major contributor of emission. Therefore, we believe that the PLQY data in mCP:TSPO1:tDPA doped film have no correlation with the photophysical properties of 2,5-tDPA:tDIDCz and 1,6-tDPA:tDIDCz.

Diphenylamine-decorated boron type MR-TADF emitters, BN2 and BN3, reported in *Adv. Funct. Mater.* 2021, 31, 2102017 also provided nearly perfect PLQY.

Fig. 1. Fluorescence emission (right axis) and excitation (left axis) spectra of (a): NDPhA ($3 \times 10^{-8} \text{ mol L}^{-1}$) and (b): DPhA ($9 \times 10^{-8} \text{ mol L}^{-1}$) before (solid line) and after (dash line) UV irradiation in water.

6. In the experimental section of device fabrication: PEDOT:PSS layer in device is produced by wet coating technique. Please specify such fabrication process as well.

Response

I appreciate your comment. We have added a detailed explanation of device fabrication process.

Added sentences

A 40 nm thick PEDOT:PSS film was embedded on ITO substrate by spin-coating (30 s at 3,200 rpm) followed by annealing at 150 °C for 15 min.

7. The operational stability result of the OLED should be provided.

Response

I appreciate your comment. We added the device operational stability of blue MR-TADF OLEDs. The lifetime test was performed at a constant current density from an initial luminance of 100 cd m^{-2} until the luminance reached 50%. However, the lifetime was short due to instability of the host and charge transport materials.

Added supporting figure

Figure S13. Luminance-lifetime curves of four MR-TADF OLEDs.

Added sentence

Figure S13 provides the device stability measured under a constant current density condition at an initial luminance of 100 cd m^{-2} . All devices exhibited half device lifetime of less than 3 h, which presumed to be due to poor material stability of host and charge transport layer.

8. Some typing problems are found in the manuscripts, for examples, “Therefore, 2,5-tDPA tDIDCz may show TADF emission, while 2,5-tDPA tDIDCz may exhibit conventional fluorescence.” Please check thoroughly.

Response

I appreciate your comment. We checked the sentences and modified them in the main text

Modified sentence

Therefore, 2,5-tDPA tDIDCz may show TADF emission, while 1,6-tDPA tDIDCz may exhibit conventional fluorescence.

REVIEWERS' COMMENTS

Reviewer #1 (Remarks to the Author):

The authors have addressed all my raised issues and made a suitable revision. I would like to recommend this manuscript for publication in its current form.

Reviewer #2 (Remarks to the Author):

can be published

Reviewer #3 (Remarks to the Author):

I appreciate the thorough clarification that addressed the concerns as raised. As seen the figures and explanation has been added accordingly, I need no further clarification in this current form.